# Effect of Carrageenans on Vegetable Jelly in Humans with Hypercholesterolemia

**DOI:** 10.3390/md18010019

**Published:** 2019-12-24

**Authors:** Ana Valado, Maria Pereira, Armando Caseiro, João P. Figueiredo, Helena Loureiro, Carla Almeida, João Cotas, Leonel Pereira

**Affiliations:** 1Polytechnic Institute of Coimbra, ESTeSC-Coimbra Health School, Department of Biomedical Laboratory Sciences, Rua 5 de Outubro, S. Martinho do Bispo, Apart. 7006, 3046-854 Coimbra, Portugal; fpereiramaria@hotmail.com (M.P.); armandocaseiro@estescoimbra.pt (A.C.); 2Marine and Environmental Sciences Centre (MARE), Faculty of Sciences and Technology, University of Coimbra, 3001-456 Coimbra, Portugal; jcotas@uc.pt (J.C.); leonel.pereira@uc.pt (L.P.); 3Unidade I&D Química-Física Molecular, Faculdade de Ciências e Tecnologia, Universidade de Coimbra, Rua Larga, 3004-535 Coimbra, Portugal; 4Polytechnic Institute of Coimbra, ESTeSC-Coimbra Health School, Department of Complementary Sciences, Rua 5 de Outubro, S. Martinho do Bispo, Apart. 7006, 3046-854 Coimbra, Portugal; jpfigueiredo@estescoimbra.pt; 5Polytechnic Institute of Coimbra, ESTeSC-Coimbra Health School, Department of Dietetics and Nutrition, Rua 5 de Outubro, S. Martinho do Bispo, Apart. 7006, 3046-854 Coimbra, Portugal; maria.loureiro@estescoimbra.pt; 6Condi Alimentar, Quinta Palmares Armazém, Rua do Ferro, 2685-459 Camarate, Portugal; carla.almeida@condi.pt; 7Department of Life Sciences, Faculty of Sciences and Technology, University of Coimbra, 3000-456 Coimbra, Portugal

**Keywords:** carrageenan, TC, HDL-C, LDL-C, TC reduction, TG

## Abstract

Changes in lipid profile constitute the main risk factor for cardiovascular diseases. Algae extracted carrageenans are long-chain polysaccharides and their ability to form gels provides for the formation of vegetable jelly. The objective was to evaluate the bioactive potential of carrageenan (E407) in the lipid profile, after ingestion of jelly. A total of 30 volunteers of both sexes, aged 20–64 years and with total cholesterol (TC) values ≥200 mg/dL, who ingested 100 mL/day of jelly for 60 days, were studied. All had two venous blood collections: before starting the jelly intake and after 60 days. At both times, TC, high density lipoprotein cholesterol (HDL-C), low density lipoprotein cholesterol (LDL-C) and triglycerides (TG), were evaluated using commercial kits and spectrophotometer. The statistics were performed using the SPSS 25.0 software and *p* < 0.05 were considered statistically significant. Serum values after 60 days of jelly intake revealed a statistically significant decrease in TC levels (5.3%; *p* = 0.001) and LDL-C concentration (5.4%; *p* = 0.048) in females. The daily intake of vegetable jelly for 60 days showed a reduction in serum TC and LDL-C levels in women, allowing us to conclude that carrageenan has bioactive potential in reducing TC concentration.

## 1. Introduction

Today, profound socioeconomic changes have led to changes in lifestyles. A sedentary lifestyle and a poor diet, low in vegetables, sometimes associated with smoking habits, are a strong inducer of alterations in lipid profile. This is evaluated in the laboratory by several parameters: total cholesterol (TC), high-density lipoprotein cholesterol (HDL-C), low-density lipoprotein cholesterol (LDL-C) and triglycerides (TG) levels. Modifications in lipid profile constitute the main risk factor in the development of cardiovascular diseases (CVDs), particularly the increase in TC and LDL-C levels [1].

Despite the description of the main risk factors and availability of pharmacological treatment, 17.9 million people die each year due to CVDs, proving to be the leading cause of death worldwide and presenting itself as one of the biggest epidemiological problems today [2]. A similar trend is observed in Portugal, with a record of 31% of deaths from CVDs [3].

According to Shimazu and collaborators, 2007, epidemiological studies have revealed a low incidence of CVDs and higher longevity in Japan, a region known for the wide introduction of algae in the diet [4]. Based on this principle, experimental and observational descriptive studies were performed on the benefits and applications of macroalgae. These studies demonstrate its richness in minerals, vitamins, dietary fiber and lipid poverty, which probably explains the epidemiological evidence [5,6].

In the diet, carrageenans are linear sulphated polysaccharides, constituents of the cell walls and intercellular spaces of *Chondrus crispus* (Rhodophyta, Gigartinales). They are extracted by aqueous or alkaline processes [7,8]. Industrially they are divided into three groups, Kappa (Κ), Lambda (λ) and Iota (ι) [7,8,9], represented on food product labels with the acronym E407 [10]. They are classified as hydrocolloids as they easily incorporate an aqueous solution leading to the formation of gels without changing the taste or color of the mixture. This feature enhances its use in the food, cosmetic and pharmaceutical industries (chocolate milk, gelatin, shampoo, and anti-inflammatories, among others) [10,11].

Carrageenans represent one of the major texturizing agents in the food industry. They are natural ingredients, which have been used for decades in food applications and regarded as safe [12]. The dairy sector accounts for a large part of the carrageenan applications in food products, such as frozen desserts, chocolate milk, cottage cheese, and whipped cream. In addition to this, carrageenans are used in various non-dairy food products, such as instant products, jellies, pet foods, and sauces, and non-food products, such as pharmaceutical formulations, cosmetics, and oil well drilling fluid [13,14]. In general, carrageenan serves as a gelling, stabilizing, and viscosity-building agent.

The literature reports multiple benefits regarding carrageenan consumption, such as, antioxidants, anticancer, antilipidemic, anticoagulants, immunomodifiers, antifungals, antivirals, and digestive health support [8,15]. In agreement with the description of the consumption of macroalgae and/or its components, such as carrageenan, its intake comes up as an alternative to the use of pharmaceutical products.

Some studies have shown an efficient reduction of serum TC, TG and LDL-C levels and an increase of HDL-C in the peripheral blood by the action of carrageenans, as a bioactive principle, mainly of Kappa subtype, justified by the interaction of its constitution (alternating groups of β (1-3)-d-galactose-4-sulfate and α (1-4)-3,6 anhydro-d-galactose) on the digestion [16,17,18]. The investigation was carried out in vitro [17] and in vivo, the latter performed mainly on Wistar rats [18,19] and population samples, where carrageenans were present as supplement [20] or in enriched foods [21].

The new concept of functional food aims to reconcile food and health benefits with the ease and speed of cooking [22,23]. In the health sciences, the possible substitution of pharmacological therapy for foods rich in bioactive and natural compounds represents a healthier and more appealing way to promote the normal functioning of the organism [24].

Carrageenans are widely used in the food industry as a thickener, gelling agent, stabilizer, and protein suspending agent [25]. Successive research and experiments on colloids led to the formation of gels with an optimal texture resulting from the interaction of carrageenan subtypes (K and ι) leading to the emergence of 100% vegetable origin jelly’s with a faster solidification [12,26,27]. The bioactive effect of the carrageenan mixture present in the vegetable jelly makes its ingestion a health benefit by reducing TC. Thus, the consumption of jellies of vegetable origin can be a healthy alternative contributing to the prevention and reduction of CVDs.

In Europe, the adherence to the Mediterranean Food Standard has been much studied to understand if the practices of individuals come close to the recommended ones. In this context, we highlight the PREDIMED instrument [28], used in a multicenter, randomized, nutritional intervention study for the primary prevention of CVDs, developed in Spain as part of the study “PREvención with DIeta MEDiterránea”, with the objective to study the effectiveness of this diet in the primary prevention of CVD. It is user-friendly in clinical setting [29], allowing rapid evaluation and immediate feedback in intervention studies.

The present investigation aimed to evaluate the impact of vegetable jelly consumption on lipid profile parameters (TC, HDL-C, LDL-C and TG) in individuals with TC levels equal to or higher than 200 mg/dL and to relate them with the degree of adherence to the Mediterranean diet.

## 2. Results

### 2.1. Sample Characterization

The experimental group of 60 days (EG-60 days), with jelly ingestion, consisted of 30 participants, 80% females and 20% males, with an average age of 48.68 ± 10.63 years. The degree of Mediterranean diet adherence was high in 50% of the individuals of this experimental group. The experimental group of 30 days (EG-30 days), with jelly ingestion, involved 12 participants with a predominance of females, 66.6% women and 33.3% men, and an average age of 44.55 ± 11.61 years. Regarding the degree of adherence to the Mediterranean diet, 8 of the 12 participants had a good adherence (Table 1).

### 2.2. FTIR-ATR Spectroscopy

The different types of jelly samples were analyzed by vibrational spectroscopy in order to confirm the existence and characterize the carrageenan type. Figure 1 shows that the four FTIR-ATR spectra of all types of jelly present the same type of spectra, with an evident particular peak of 1065 cm^−1^. This indicates a gelling type of carrageenan (DA) [9], verified in the kappa and iota carrageenans, which have inherent gelling properties [30].

All the spectra showed a band at 849 cm^−1^, which is characteristic of D-galactose-4-sulfate (G4S) present in the kappa and iota carrageenan types.

The peak at 908 cm^−1^ indicates the presence of 3,6-anydrogalactose-2-sulfate (DA2S), a type of linkage that belongs to the iota carrageenan type. The peak at 943 cm^−1^ indicates the C–O–C of 3,6-anhydrogalactose (DA) linkage, which is related with the presence kappa or iota carrageenan in the sample. The band at 987 cm^−1^ shows the presence of Galactose (G/D) typically present in the kappa and/or iota types of carrageenan. Nonetheless, the peaks at 1005 cm^−1^, 1049 cm^−1^, 1115 cm^−1^, 1126 cm^−1^, and 1134 cm^−1^ indicate the presence of sulfate groups, such as sulfate esters and sulfated galactans.

Concluding, vegetal jellies present the same spectra for four different flavors indicating that all flavors have the same colloid, a hybrid kappa/iota carrageenan which is typically present in jellies from vegetal origin (Figure 1) [31].

### 2.3. Total Cholesterol

The total cholesterol parameter, in the group that ingested carrageenan for 30 days (EG-30 days) at T0 had a concentration of 244.86 ± 31.94 mg/dL and at T1 of 226.18 ± 40.99 mg/dL, representing a statistically significant decrease of 7.63% (18.67 mg/dL), *p* = 0.021 (Figure 2). The group of individuals evaluated after 60 days of ingestion (EG-60 days) presented TC concentration at T0 of 236.92 ± 30.48 mg/dL and at T2 of 224.40 ± 23.47 mg/dL, revealing a statistically significant decrease of 5.3% (12.52 mg/dL), *p* = 0.001 (Figure 3). Analysis by gender showed a significant decrease (*p* = 0.004) in women at T2, comparatively to T0 (Table 2).

### 2.4. HDL-Cholesterol

The EG-30 days group, before ingestion of the jelly (T0), had an average concentration of 62.20 ± 19.30 mg/dL. After 30 days (T1), it presented 56.88 ± 17.39 mg/dL, which represents a statistically significant decrease of 8.6% (5.32 mg/dL), *p* = 0.008 (Figure 2). The concentration of HDL-C in EG-60 days presented at T0 a concentration of 57.60 ± 13.30 mg/dL and at T2 of 54.97 ± 12.78 mg/dL, with a decrease of 4.6% (2.63 mg/dL) in the parameter, with statistical significance, *p* = 0.037 (see Figure 3). The distribution by gender revealed no statistically significant differences and is shown in Table 2.

### 2.5. LDL-Cholesterol

The LDL-C parameter concentration at EG-30 days presented mean values of 209.54 ± 57.13 mg/dL at T0 and at T1 of 208.87 ± 64.36 mg/dL, showing no differences (Figure 2). The individuals belonging to the EG-60 days presented a mean T0 concentration of 197.39 ± 36.5 mg/dL and T2 of 189.83 ± 33.73 mg/dL, showing a 3.8% decrease (7.56 mg/dL; *p* = 0.062) (Figure 3). Considering an evaluation by gender of the individuals in the statistical analysis of the parameter, a statistically significant reduction of 5.4% (10.28 mg/dL) was obtained for females, *p* = 0.048 (Table 2).

### 2.6. Triglycerides

The EG-30 days exhibited at T0 a concentration of 116.09 ± 30.02 mg/dL and at T1 a concentration of 151.28 ± 31.8 mg/dL, representing a 30.3% increase (35.19 mg/dL), statistically significant, *p* = 0.038 (Figure 2). The EG-60 days group presented at T0 the serum concentration of 92.93 ± 38.88 mg/dL and at 60 days (T2) the average concentration was 109.55 ± 38.35 mg/dL, representing a statistically significant increase of 17.8% (16 mg/dL), *p* = 0.014 (Figure 3). Analysis by gender revealed a significant increase associated with males at time T2, versus T0 (Table 2).

### 2.7. MeDiet-Adherence to Mediterranean Diet

The analysis of the results considering the MeDiet questionnaire evaluation led to the conclusion that the adhesion or not of a Mediterranean diet does not influence the action of carrageenan, present in jelly, in the lipid profile.

## 3. Discussion

Changes in lipid profile constitute the main risk factor in the development of CVDs, considered the leading cause of death worldwide [1,2]. The need to reduce the mortality rate has led to epidemiological studies showing that seaweed ingestion induces an antilipidemic effect [4,5]. Intake of fiber-rich foods has been described as a protective factor for numerous pathologies, such as hypertension, obesity, diabetes, and many others [32]. Algae, rich in water-soluble fibers such as carrageenan, play an important role as a source of fiber in the diet [13,22]. Introduction of seaweed constituents in food products is nowadays frequently and widely used. As example, the inclusion of Kappa–Iota hybrid carrageenan, identified by food additive code E407, does not alter the taste or color of mixtures and optimizes their consistency, texture, and water retention [7,8]. Due to this characteristic, carrageenan is widely used as a gelling agent in the food industry in the production of chocolate milk, ice cream, beer, and jelly [10]. In addition to the antilipidemic effect, numerous benefits to human health are attributed to carrageenan as antioxidant, immunomodulatory, antiviral, and digestive health support, thus revealing its high bioactive potential [9,14]. However, for cultural reasons it is difficult to estimate the impact of a carrageenan-rich diet associated with individuals not inserted in populations of Asian culture and to evaluate their lipid profile, giving the present study a pioneer status. Thus, two time periods of 30 and 60 days of daily ingestion of vegetable origin jelly were tested. In the absence of studies with similar methodology developed in Caucasian populations, the time periods were based on studies by Sokolova et al. [20] and by Panlasigui et al. [21].

According to Panlasigui and coauthors, participants ingested the mixed carrageenans in their own food for eight weeks [21]. Also, Sokolova’s research applied carrageenan as a dietary supplement to individuals previously diagnosed with CVDs [20]. Participants ingested 250 mL per day of a supplement consisting of a hybrid K–λ carrageenan (3:1) for 20 days [20]. Our results showed a statistically significant decrease in TC and HDL-C concentration in both periods (60 and/or 30 days) after daily ingestion of 100 mL of vegetable jelly. Also, a reduction in TC and LDL-C levels in females when assessed by gender distribution and EG-60 days was noted. In contrast, males recorded an increase in TG concentration.

The TC levels decreased by 7.6% after 30 days and 5.3% after 60 days. The decrease was corroborated by Sokolova et al. [20], but with a smaller percentage, which may be due to the different study design involving cardiovascular patients with supplemental carrageenan consumption resulting in a 16.5% decrease in TC [20]. Also, Panlasigui and coauthors [21], in their research on eating carrageenan-rich foods at the three most important meals (breakfast, lunch and dinner) showed a reduction in TC levels by 33%. Concerning HDL-C concentration, our results decreased by 8.6% and 4.6% after 30 and 60 days respectively, as opposed to the work of other authors [20,21] which showed an increase. For the concentration of the LDL-C fraction after 60 days, it showed a tendency to decrease, evidenced in relation to females showing a significant decrease, which is in accordance with the results of Panlasigui et al. and Matthan et al. [21,33]. The increase in TG concentration was 30.3% for 30 days and 17.8% after 60 days of jelly ingestion. It should be noted that it has not reached the maximum reference values [34]. For TG, the literature presents ambiguous results, since Panlasigui and coauthors showed a 32% reduction [21], while Sokolova and collaborators showed an increase [20]. However, low cholesterol absorption is known to be associated with increased TG and decreased HDL in both sexes [33]. Despite the multiple benefits to human health, the mechanism of action of carrageenans is not yet completely understood and only their interaction with the digestive process is documented, being classified as long-chain carbohydrates [35,36]. When carrageenans reach the intestinal tract, the volume and viscosity of the intestinal content increases and leads to increased digestion time and reduced efficiency, since the availability of products for enzymatic action is compromised [37].

On the other hand, it is capable of capturing cholesterol and bile salts, inhibiting lipase, causing a decrease in cholesterol absorption directly and indirectly, since the emulsifying action performed by bile salts and lipase becomes deficient, disabling micelle formation [38,39,40]. However, the action of carrageenans is not restricted to the partial effect of gastrointestinal barrier, but also has endogenous repercussions. Excretion of bile salts precludes their reuse and, therefore, synthesis is required once more [20,21,40,41]. The mobilization of TG occurs from energy reserves into the bloodstream because cholesterol is required for synthesis and it is in low circulating concentrations [42].

The bioactive potential of carrageenan comes from its ability to decrease cholesterol absorption, leading to a decreased cholesterol absorption rate and consequent increase in endogenous cholesterol synthesis rate, maintaining the balance of values, a fundamental process to maintaining the proper functioning of the body [33]. According to some researchers, the relationship between cholesterol absorption markers and serum parameters has shown that when cholesterol absorption markers decrease, serum triglyceride concentration increases and HDL-C levels decrease [33,43]. As a complement, Matthan et al. [33] verified that when high endogenous cholesterol level marker values were detected, the serum LDL-C fraction values decreased. An inverse relationship between expression markers and LDL-C concentration is justified by the increased expression of the LDL-C receptor in the cell, which causes an increase in cholesterol internalization rate, thereby decreasing 3-hydroxy-3-methylglutaryl-coenzyme A (HMG-CoA) reductase activity and, therefore, decreasing of endogenous cholesterol synthesis [33,43]. Thus, carrageenan ingestion leads to decreased cholesterol absorption rate, leading to increased serum TG concentration and reduced HDL-C. On the other hand, the need for cholesterol for bile salt synthesis, as a result of carrageenan-induced excretion, leads to the promotion of endogenous cholesterol synthesis, leading to a decrease in serum LDL-C levels [20,40,41]. Also, the gender difference leads to metabolic changes where the hypolipidemic properties of estrogens induce a protective role in women [44]. The low activity of the enzyme HMG-CoA reductase in women indicates that the neo-synthesis of cholesterol is physiologically lower in females, until menopause, compared with males [33,44]. Also, adding the principle that the presence of estrogens increases intestinal cholesterol absorption in humans, it seems to justify the significant reduction observed in LDL-C and TC levels in women as a result of decreased absorption efficiency of induced cholesterol molecules by vegetable jelly [41,44].

In conclusion, carrageenans constitute a bioactive potential in reducing total cholesterol levels in the body. Regular consumption of vegetable origin jelly is beneficial, contributing to lower total cholesterol levels and also to decreased LDL-C levels in females. Ingestion of vegetable jelly can therefore be considered a good practice as a protection against the development of cardiovascular diseases.

Although these results are considered relevant, the need to increase the number of participants in this type of study in order to consolidate the presented results in future complementary works must be emphasized.

## 4. Materials and Methods

### 4.1. Sample Characterization

The study comprised 30 volunteers of both sexes, aged between 20 and 64 years. In the selection of the sample, the following inclusion criteria were applied: over 18 years old, with previous indication of elevated TC levels (equal to or higher than 200 mg/dL), and apparently healthy. No restriction on participants’ eating habits was indicated. As exclusion criteria, pharmacological therapy for hypercholesterolemia was considered.

The study participation involved the ingestion of 100 mL of vegetable jelly per day, preferably after dinner, for 60 days. This feature allowed the formation of two experimental groups. The experimental group with jelly ingestion for 60 days (EG-60 days) was composed of 30 participants. Of these, 12 were harvested after 30 days, constituting the experimental group with jelly ingestion for 30 days (EG-30 days). In the jelly composition, the presence of a hybrid Kappa–Iota carrageenan polysaccharide, integrated in the gelling agent, was equal in the various flavours (pineapple, green tea, mango, strawberry). The analysis of the carrageenan composition present in the jellies was performed by attenuated total reflectance Fourier transform infrared spectroscopy (FTIR-ATR) spectroscopy on the IFS 55 equipment using the Golden Gate ATR single reflection diamond system. The jelly with vegetable origin is available on any commercial food surface. In its composition we find the following ingredients: sugar, gelling agent (carrageenan, potassium citrates and sucrose), acidity regulators (citric acid and sodium citrates), aromas (sulfites), antioxidant (ascorbic acid, vitamin C) and dyes (anthocyanins and beta carotene). For the composition and energy distribution of a portion of 92 g, equivalent to 100 mL of jelly ingested, the following mean values are given: energy 266 kJ/63 kcal (3% DR) where DR is the reference dose for an average adult (8400 kJ/2000 kcal) per serving; lipids, including saturated 0 g; carbohydrates 16 g (6% DR), of which 15 g (17% DR) are sugars; fiber 0 g; proteins 0 g; salt 0.08 g (1% DR); Vitamin C: 29 mg (36% NRV) where NRV is the nutrient reference value. The preparation of jelly involves the addition of a 90 g sachet of powdered preparation to 500 mL of boiling water, followed by distribution into five portions.

According to the inclusion criteria, venous blood was collected at three different times: (T0) before starting the consumption of vegetable jelly, (T1) after 30 days and (T2) 60 days after the first jelly. Posteriorly, the analytical parameters TC, HDL-C, LDL-C and TG were performed at various times and the adherence to the Mediterranean diet was evaluated.

### 4.2. Sample Collection and Preparation

Venous blood samples were collected (about 5 mL) into a vacuum-dried gel tube and centrifuged at 1800× *g*, 4 °C for 10 min. Serum was aliquoted and stored at −20 °C until adjustments were made. The TC was quantified by the Liquick cor-CHOL KIT, from Cormay, Poland, according to the enzymatic colorimetric glycerophosphate oxidase method. The determination of TG was performed by the Liquick cor-TG KIT, from Cormay, Poland, using the enzymatic colorimetric method with cholesterol esterase and cholesterol oxidase. The HDL-C quantification was performed with the Prestige 24i HDL-DIRECT KIT, Cormay, Poland with help of the enzymatic direct colorimetric method using cholesterol esterase and cholesterol oxidase. The concentrations were evaluated by spectrophotometry according to the protocol and the Prestige 24i equipment, *Tokyo Boeki Medical System* Ltd., Tokyo, Japan.

The determination of LDL-C was calculated by Friedewald’s formula (LDL-C = [TC]–[HDL-C]–([TG]/5)) using the TC, HDL-C and TG concentrations previously quantified. The formula is applied only at TG concentrations <400 mg/dL [34,45].

The adherence to the Mediterranean diet was evaluated by applying the MeDiet questionnaire, integrated in PREDIMED [28]. Demographic characterization was also obtained by the questionnaire. The PREDIMED questionnaire presents 14 questions from which the respondent is categorized as having good or poor adherence to Mediterranean diet. The response to each of the 14 items is scored 1 if it meets the criteria defined as characteristic of this type of feed (possible range 0–14 points). A final score of 10 points or higher represents good adherence to the standard. This questionnaire uses predefined targets for food consumption [46].

### 4.3. Statistical Analysis

The statistical treatment was performed using SPSS 25.0 Software, IBM^®^, United States and the graphs using GraphPad Prism version 8.1.2. Descriptive analysis of the variables was performed with mean, maximum, minimum, and standard deviation calculations. The Wilcoxon’s t-test was used for comparison of paired quantitative variables. The results are presented as mean ± standard deviation and considered statistically significant for *p*-value < 0.05.

### 4.4. Confidentiality and Data Protection

The study was conducted in accordance with the principles of the Declaration of Helsinki, ensuring maximum protection and confidentiality of data, having been previously approved by the Ethics Committee of the Polytechnic Institute of Coimbra. All subjects gave their informed consent for inclusion before they participated in the study. The study was conducted in accordance with the Declaration of Helsinki, and the protocol was approved by the Ethics Committee of Polytechnic Institute of Coimbra (nº 44/2018).

## Figures and Tables

**Figure 1 marinedrugs-18-00019-f001:**
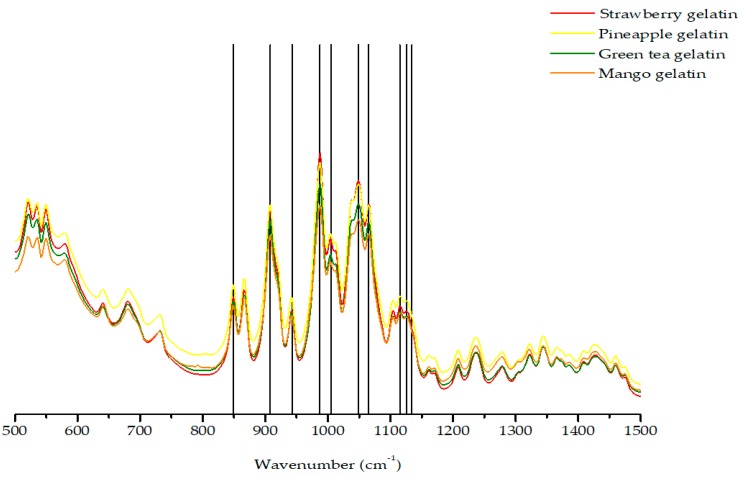
The four FTIR-ATR spectra of different jelly flavors present the same type of spectra with an evident peak of 1065 cm^−1^ that indicates kappa (κ) and iota (ι) hybrid carrageenan [9]. Vegetal jellies present the same spectra indicating they have the same colloid, a hybrid kappa/iota carrageenan.

**Figure 2 marinedrugs-18-00019-f002:**
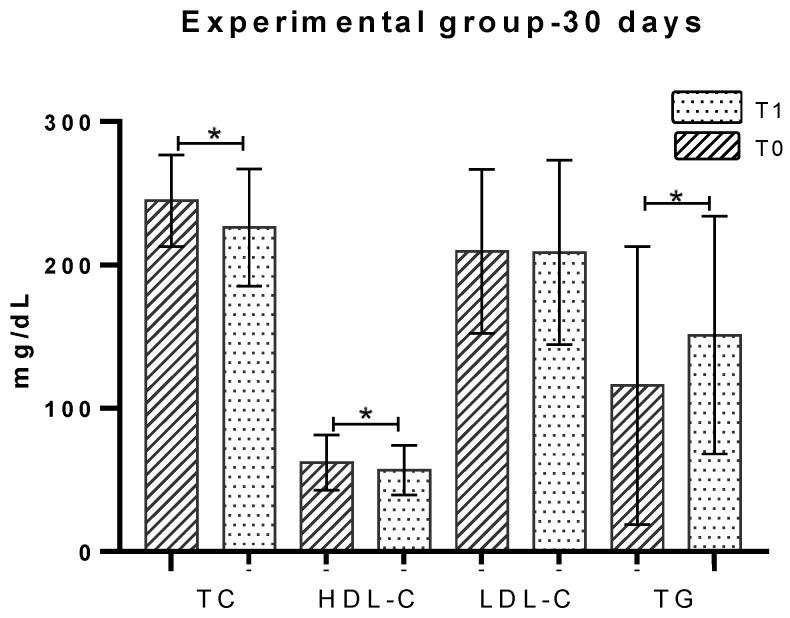
Lipid profile evaluation in experimental group with vegetable jelly ingestion for 30 days. T0: before consumption of jelly; T1: after 30 days of consumption of jelly; TC: Total cholesterol; HDL-C: High-density lipoprotein cholesterol; LDL-C: Low-density lipoprotein cholesterol; TG: Triglycerides; * *p* < 0.05.

**Figure 3 marinedrugs-18-00019-f003:**
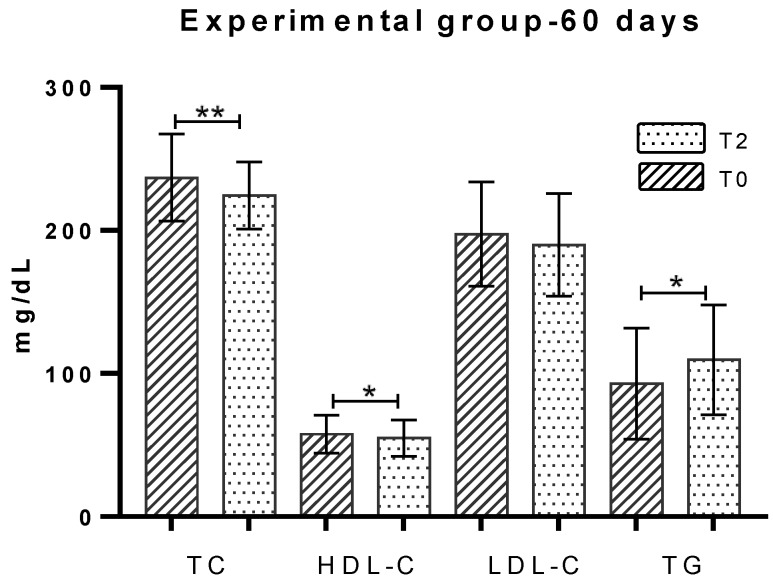
Lipid profile evaluation in experimental group with vegetable jelly ingestion for 60 days. T0: before consumption of jelly; T2: after 60 days of consumption of jelly; TC: Total cholesterol; HDL-C: High-density lipoprotein cholesterol; LDL-C: Low-density lipoprotein cholesterol; TG: Triglycerides; * *p* < 0.05; ** *p* < 0.01.

**Table 1 marinedrugs-18-00019-t001:** Sample characterization.

Ingestion Period		30 Days	60 Days
		*n* = 12	*n* = 30
Age (years)	-	44.55 ± 11.61	48.68 ± 10.63
(min–max)		(23–64)	(20–64)
Sex	F	8	24
	M	4	6
MeDiet	≥10	4	15
	<10	8	15

F: Female; M: Male; MeDiet: Mediterranean diet.

**Table 2 marinedrugs-18-00019-t002:** Characterization of lipid profile analytical parameters by sex on EG-60 days.

	Sex	
	Female	Male
TC (T0)	234.51 ± 31.06	244.95 ± 29.62
TC (T2)	220.43 ± 21.16 **	237.76 ± 27.82
HDL-C (T0)	60.85 ± 13.15	46.78 ± 6.65
HDL-C (T2)	58.41 ± 12.10	43.46 ± 7.28
LDL-C (T0)	190.58 ± 33.85	220.12 ± 38.77
LDL-C (T2)	180.35 ± 29.07 *	221.48 ± 40.11
TG (T0)	87.88 ± 38.06	109.75 ± 40.18
TG (T2)	101.65 ± 31.16	135.87 ± 50.83 *

The results are presented in mg/dL. TC: Total cholesterol; HDL-C: High-density lipoprotein cholesterol; LDL-C: Low-density lipoprotein cholesterol; TG: Triglycerides. T0: before consumption of jelly; T2: after 60 days of consumption of jelly; * *p* < 0.05; ** *p* < 0.01 versus T0.

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
