# Peer review of "Effect of Carrageenans on Vegetable Jelly in Humans with Hypercholesterolemia"

_marinedrugs, 2019, doi:10.3390/md18010019_

Round 1

Reviewer 1 Report

The study explores the effect of Carrageenans extracted from seaweeds on humans with Hypercholesterolemia. Although the study does not bring any significant novelty, it is well written and well approached with the literature. The authors clearly described the objective of the study, and the added value it can bring. The structure of the manuscript needs some adjustment given that some mistakes were detected. The English level is very good. The manuscript and the study fit the scientific approach of the journal and, therefore, the study can be accepted for publications in the Marine Drugs journal, but not in its present form because some adjustments are required. Below are my comments and remarks:  

Table 1 at the end of the introduction should be next to its legend in section 2.1 of the Results. The text of section 2.3 is all centered, therefore the authors need to justify the text. Some information presented in section 2.3 belongs to section 2.4. The authors are required to re-arrange the text and distinguish well between the information that belong to each section. The sentence right above Figure 2 belongs to the legend of Figure 2 and not to the text of section 2.3. The legend of figure 3 should be right underneath the figure and not after the text. Some journal abbreviations in the reference list are wrongly abbreviated, the authors need to double-check the abbreviations of the reference list and correct them.   The authors mentioned many times that a significant decrease in LDL-C concentration in females. However, there is no clear scientific explanation on why this is happening for females and not males, why?. I suggest that the authors elaborate more on this question. The authors mentioned the TC level decreases including HDL-C, but then they mention that the TG level increases in the serum. So what is the benefit of using a method to decrease TC that will create another problem for the patients by increasing the TG level? Knowing that high TG levels can lead to stroke, heart disease, heart attacks, and other chronic diseases.  The second paragraph of section 4.3 (statistical analysis) is indeed very important. But it has to be split from section 4.3 and put in a new section 4.4 called "Confidentiality and data protection". Line 202 - 203: To avoid confusing the readers, the authors need to mention that the level of LDL-C did not change after EG-30 days, as can be seen in Figure 3.  Where are the data that distinguish between females and males? Given that the authors mention several times in the manuscript that LDL-C significantly decreases for females, but the data are not displayed in the manuscript. It will make more sense to put section 2.4 before section 2.3 because the study starts with EG-30 days, then EG-60 days. 

Author Response

The study explores the effect of Carrageenans extracted from seaweeds on humans with Hypercholesterolemia. Although the study does not bring any significant novelty, it is well written and well approached with the literature. The authors clearly described the objective of the study, and the added value it can bring. The structure of the manuscript needs some adjustment given that some mistakes were detected. The English level is very good. The manuscript and the study fit the scientific approach of the journal and, therefore, the study can be accepted for publications in the Marine Drugs journal, but not in its present form because some adjustments are required. Below are my comments and remarks:  

Table 1 at the end of the introduction should be next to its legend in section 2.1 of the Results. The text of section 2.3 is all centered, therefore the authors need to justify the text. Some information presented in section 2.3 belongs to section 2.4. The authors are required to re-arrange the text and distinguish well between the information that belong to each section. The sentence right above Figure 2 belongs to the legend of Figure 2 and not to the text of section 2.3. The legend of figure 3 should be right underneath the figure and not after the text.

We accepted reviewer suggestions and the text and text formatting were adjusted.

 Some journal abbreviations in the reference list are wrongly abbreviated, the authors need to double-check the abbreviations of the reference list and correct them.  

– We followed reviewer suggestion and the abbreviations were corrected.

The authors mentioned many times that a significant decrease in LDL-C concentration in females. However, there is no clear scientific explanation on why this is happening for females and not males, why?. I suggest that the authors elaborate more on this question. The authors mentioned the TC level decreases including HDL-C, but then they mention that the TG level increases in the serum. So what is the benefit of using a method to decrease TC that will create another problem for the patients by increasing the TG level? Knowing that high TG levels can lead to stroke, heart disease, heart attacks, and other chronic diseases. 

We understand reviewer concern in respect to TG levels observed, but TG levels keep under the reference values for low risk (<150 mg/dl) at EG-60, as referred in reference 34.

The second paragraph of section 4.3 (statistical analysis) is indeed very important. But it has to be split from section 4.3 and put in a new section 4.4 called "Confidentiality and data protection". Line 202 - 203:

We followed reviewer suggestion and the text position in the document has been adjusted.

To avoid confusing the readers, the authors need to mention that the level of LDL-C did not change after EG-30 days, as can be seen in Figure 3. 

We agree with reviewer and in the document is described (line 183) that we didn’t find differences at EG-30 days in LDL-C levels.

Where are the data that distinguish between females and males? Given that the authors mention several times in the manuscript that LDL-C significantly decreases for females, but the data are not displayed in the manuscript.

We followed reviewer suggestion and table 2 were added to the document with the LDL-C levels according sex in EG-60.

It will make more sense to put section 2.4 before section 2.3 because the study starts with EG-30 days, then EG-60 days. 

We followed reviewer suggestion and the order of the text was changed.

Reviewer 2 Report

The manuscript by Ana Valado et al., reports the effect of vegetable jelly consumption on lipid profile parameters (TC, HDL-C, LDL-C, and TG) in humans. The authors showed that the carrageenans comprise a bioactive component which is potential in reducing total cholesterol levels in the body, especially to decrease LDL-C levels in females. Their results suggested that the Ingestion of vegetable jelly can be considered a good practice as a protection against the development of cardiovascular diseases.

Overall, the manuscript was written well. I would recommend the publication of this manuscript after the following minor revisions.

The authors used 80% female and 20 % male and the results have shown that a reduction in serum TC and LDL-C levels in the females. Applying the results of females, the authors concluded that carrageenan has bioactive potential in reducing TC concentration. There is a big gap to conclude the results to evaluate only females. The authors should be shown the results between males and females. In a separate figure, the results should be presented to compare their potentiality. In the discussion section, the authors should discuss the results of females with relevant references and why the reduction of cholesterol levels only on the females, why not on the males.   

Author Response

Overall, the manuscript was written well. I would recommend the publication of this manuscript after the following minor revisions.

The authors used 80% female and 20 % male and the results have shown that a reduction in serum TC and LDL-C levels in the females. Applying the results of females, the authors concluded that carrageenan has bioactive potential in reducing TC concentration. There is a big gap to conclude the results to evaluate only females. The authors should be shown the results between males and females. In a separate figure, the results should be presented to compare their potentiality. In the discussion section, the authors should discuss the results of females with relevant references and why the reduction of cholesterol levels only on the females, why not on the males.   

We followed reviewer suggestion and a table (nº2) with the results for EG-60 by sex were included as well as the discussion about the influence of sex in these results.

Round 2

Reviewer 1 Report

The authors took into consideration my comments and answered my questions adequately. The paper can be accepted for publication

Author Response

We acknowledge the comments and the aceptation of the last version of document.

Reviewer 2 Report

Still, there is a gap to conclude the results between the male and females. As I mentioned in my previous review, the authors should be shown the results clearly between males and females. Experimental results should be presented to compare the activity in terms of cholesterol reduction. I am convinced with the discussion of females with relevant references and why the reduction of cholesterol levels only on the females, why not on the males. However, a comparison of the results on the basis of the experiment and literature is essential to discuss clearly for the readers.

Author Response

Dear Editor

The authors accepted the comments of the reviewers and responded accordingly. In the first review we found that, much of the manuscript was deformed. Clarified the doubts, first reviewer was accepted the revised manuscript. The second reviewer, with some questions, proposed a second revision, suggesting the inclusion of additional results. The authors added an additional table (table 2) with the results and the respective discussion.
As the manuscript of the first revision also showed changes, the authors, as a safeguard, submit revision 2 of the manuscript with blue (first revision) and green (second revision) text.

Best regards.

Ana Valado

Responses to reviewer 2

Reviewer 2: Still, there is a gap to conclude the results between the male and females. As I mentioned in my previous review, the authors should be shown the results clearly between males and females. Experimental results should be presented to compare the activity in terms of cholesterol reduction. I am convinced with the discussion of females with relevant references and why the reduction of cholesterol levels only on the females, why not on the males. However, a comparison of the results on the basis of the experiment and literature is essential to discuss clearly for the readers.

Authors responses: The authors accept the comments and acknowledge the importance of the results, but understood only with regard to LDL-C levels for both reviewers. The results are presented in full and by sex in Table 2. There was a reduction in total cholesterol that was discussed and referenced in the discussion. The results are most evident in women due to its physiology. And, as pointed out, it is important to expand the study in future for more robust results.
The corrections made in the manuscript regarding the questions posed by the reviewer 2 are written  green.
